# Impacts on Context Aware Systems in Evidence-Based Health Informatics: A Review

**DOI:** 10.3390/healthcare10040685

**Published:** 2022-04-05

**Authors:** Stella C. Christopoulou

**Affiliations:** Department of Business Administration and Organizations, University of Peloponnese, Antikalamos, 24100 Kalamata, Greece; s.xristopoulou@uop.gr

**Keywords:** smart environment, smart space, context-awareness, pervasive technologies, *Evidence-Based Health Informatics* (*EBHI*), Health Informatics, mobile applications, review

## Abstract

Background: The application of *Context Aware Computing* (*CAC*) can be an effective, useful, feasible, and acceptable way to advance medical research and provide health services. Methods: This review was conducted in accordance with the principles of the development of a mixed methods review and existing knowledge in the field via the *Synthesis Framework for the Assessment of Health Information Technology* to evaluate *CAC* implemented by *Evidence-Based Health Informatics* (*EBHI*). A systematic search of the literature was performed during 18 November 2021–22 January 2022 in *Cochrane Library*, *IEEE Xplore*, *PUBMED*, *Scopus* and in the clinical registry platform *Clinicaltrials.gov*. The author included the articles in the review if they were implemented by *EBHI* and concerned with *CAC* technologies. Results: 29 articles met the inclusion criteria and refer to 26 trials published between 2011 and 2022. The author noticed improvements in healthcare provision using *EBHI* in the findings of *CAC* application. She also confirmed that *CAC* systems are a valuable and reliable method in health care provision. Conclusions: The use of *CAC* systems in healthcare is a promising new area of research and development. The author presented that the evaluation of *CAC* systems in *EBHI* presents positive effects on the state of health and the management of long-term diseases. These implications are presented in this article in a detailed, clear, and reliable manner.

## 1. Introduction

Along with the increasing efforts for efficiency and safety in the provision of health care, the interest in *HIT* (*Health Information Technologies*) is constantly increasing.

According to the US National Library of Medicine’s *MeSH* (http://www.nlm.nih.gov/mesh/MBrowser.html, accessed on 30 March 2022) Concept Dictionary, the term *HIT* is usually described in three terms: (i) *Medical Informatics Applications*, (ii) *Health Information Exchange*, and (iii) *Computational Medical Informatics* (either *Medical Informatics Computing*).

The term *Medical Informatics Applications* (https://meshb.nlm.nih.gov/record/ui?ui=D008491, accessed on 30 March 2022) refers to the automated systems applied to the patient care process including diagnosis, therapy, and systems of communicating medical data within the health care setting.

The term *Health Information Exchange* (https://meshb.nlm.nih.gov/record/ui?ui=D066275, accessed on 30 March 2022) refers to the organizational framework for the dissemination of electronic healthcare information or clinical data, across health-related institutions and systems.

The term *Computational Medical Informatics* (either *Medical Informatics Computing*) (https://meshb.nlm.nih.gov/record/ui?ui=D008492, accessed on 30 March 2022) is used to describe the precise procedural mathematical and logical operations utilized in the study of medical information pertaining to health care.

Also the terms *eHealth*, *telemedicine*, and *mHealth* (https://meshb.nlm.nih.gov/record/ui?ui=D017216, accessed on 30 March 2022), although often used interchangeably, each of them has a separate definition and content. All three terms, however, are used to describe the combined use of technologies and communications to provide health services anywhere, anytime [1]. *Tele-medicine* includes the provision of health services through remote telecommunications and supports interactive counseling and diagnostic services. Moreover, mobile health is considered as a medical and public health practice supported by mobile devices such as smartphones, personal digital aids, and sensors [2].

The key technical issues for the proper operation of *HITs* are to ensure on the one hand their reliability and validity, to serve and satisfy the medical community, and on the other to provide safety and ease of use by users-patients and medical users. Cost-effective solutions are an additional criterion, if the long-term benefits of improving public health and saving resources, both at the private and public levels, are considered.

However, the most essential question that arises is whether and to what extent these technologies are integrated in an effective, integrated, and safe way for the patient. The answer is that this is achieved in the most reliable way through the *EBM* (*Evidence-Based Medicine*) application.

*EBM* is defined as a thorough and rational synthesis of the best evidence for patient care decision-making [3]. The field of *EBM* emerged more than thirty years ago and since then it is constantly developing following the need of the medical community to apply reliable guidelines in daily medical and nursing practice. *EBM* results from the synthesis of clinical experience, evidence, and personal assessment of the patient. Evidence can be derived both from existing medical research and literature as well as from data from the practical medical application [4]. The purpose of *EBM* is to apply the best available data from the current and applied scientific methodology and literature for medical decision-making but also to evaluate the quality of the available data regarding the risks and benefits of treatment [5]. The bibliographic sources related to *EBM* include clinical trials and mostly *RCTs* (*Randomized Controlled Trials*), systematic reviews, Clinical Practice Guidelines, quasi-Experimental studies, descriptive studies, and expert opinions [6].

To support *EBM*, the *RCTs* is the most common well-designed type of invasive clinical trial application, which aim to compare and evaluate diagnostic and therapeutic strategies, methods, and tools. The research and medical community have widely accepted the conduct of *RCTs* and considers this to be the ‘gold standard’ [3], second in the ranking after the systematic review. This is because *RCTs* provide a strict standard that clearly defines the design, conduct, monitoring, control, recording, analysis, and reporting of clinical trials to ensure that data and outputs are reliable and accurate. Alongside, their implementation ensures the rights, integrity, and confidentiality of those involved [3,7].

The area that supports health information technology platforms and *e-health* interventions in conjunction with clinical trials is called *EBHI* (*Evidence-Based Health Informatics*) [8]. *EBHI* is based on the principles of *EBM* [3] in conjunction with the implementation of *HIT*. Therefore, *EBHI* is defined as the thorough and rational synthesis of optimal evidence (i.e., as evidenced by the implementation of reliable clinical trials) that is valid for decision-making on the introduction and operation of *HIT* [9]. *EBHI* is currently at the forefront of physicians’ support in clinical decision-making. By adopting it, those who design, develop, and implement health information systems rely not only on science or experts, but also on clear and accurate data from rigorous and up-to-date studies. These studies analyze in detail what makes these systems clinically acceptable safe and effective.

However, in addition to examining current trends in terms of health, the combination of *EBHI* with modern technologies is extremely interesting and provides many positive impacts. As well, *Context Aware Computing* (*CAC*) is one of the most modern and innovative technologies. More specifically, *CAC* is a challenge of distributed mobile computing aimed at managing next-generation smart applications where personalized devices (sensors, biosensors, mobile phones, iPads, etc.) interact with users to create a smart environment. A *context* is that which surrounds the user or more generally the entity under study (e.g., devices, applications). This term is used in relation to the physical world that surrounds a device, application, or system. Thus, *CAC* covers a wide range of applications that can perceive their environment and react intelligently according to this perception. Such a system can detect and react to changes in the environment.

There are three important aspects of a context-aware environment: (a) where the user (or the entity in general) is located, (b) what entities/objects are around it, and (c) what are its neighboring sources of information.

The context (environment) may include in addition to the user’s location, lighting, noise level, network interface, communication costs, communication bandwidth, and anything else that is considered useful for the user such as his state of health, his safety, or even his social condition. e.g., that he is with the manager or an associate [10].

Especially in relation to telemedicine and the continuous and personalized provision of counseling to the patient from everywhere a *CAC* system can obtain and translate the relevant information of the environment and other inherent factors (system logic e.g., applicable medical rules and reasoning) and perform the necessary actions to provide care.

*CAC* is modeled based on a set of interrelated parameters and their perception. These parameters, the so-called environment parameters, are categorized into the following categories: (a) User and Role: This category provides a categorization of users according to their roles, such as diverse types of customers or several types of employees. (b) Process and Task: This category represents a functional framework, such as work data for employees. (c) Location: It concerns the categorization of application-related sites, with the desired detail: for some applications, the country may be sufficient location information, for others, the city and so on. (d) Time: This category refers to different types of time information, such as customer time zone, real-time, virtual time, etc., and (e) Device: This category contains information about the device that supports the system to provide relevant environmental or vital parameters [11].

The *CAC* systems are classified into the following categories in terms of design of environmental intelligence systems (ambient intelligence) and framework awareness systems for home care models in systems that support (a) emergency treatment (i.e., services for emergency detection and management) (b) strengthening autonomy, i.e., customer/patient support services to meet their basic needs and/or their daily activities (e.g., diet, medication, monitoring of vital signs, etc.) and (c) comfort services, i.e., services that promote a better quality of life for the patient (e.g., education, socialization, etc.) [12].

Blending the above, the combined application of *EBHI* and *CAC* systems will be studied below.

### Objectives

The aim of this study was to review the existing literature on the impacts of health-related interventions using context-aware technologies implemented with the support of *EBHI*. It is estimated that the research in the literature described studies that are controlled by clinical trials provide more complete and more reliable results.

Thus, the author identified three broad *Research Questions* (*RQs*) that will guide the rest of this work.

-*RQ1*: What are the most common categories of applied information science using *CAC* systems found in the literature that were evaluated via *EBHI*?-*RQ2*: What are the most common categories of medical/health applications using *CAC* systems found in the literature that were evaluated via *EBHI*? More specifically, how disease-specific factors mediate the impact of *CAC* systems in *EBHI*?-*RQ3*: What are the impacts of the studies/trials using *CAC* systems related with a medical/health domain and *HITs* found in the literature that were evaluated via *EBHI*? Moreover, when are *CAC* systems and in which conditions are they considered the most acceptable and satisfactory solutions by users?

## 2. Materials and Methods

### 2.1. Search Strategy

The author performed a systematic search of the literature from 18 November 2021–22 January 2022 in *Cochrane Library*, *IEEE Xplore*, *PUBMED*, *Scopus*, and in the clinical registry platform *Clinicaltrials.gov* using search keywords regarding the terms: (*smart* OR *pervasive* OR *context aware*) AND (*system* OR *environment* OR *application* OR *device* OR *place* OR *space* OR *home*) AND (*health* OR *medical* OR *medicine*) AND *trial*.

The author also screened the reference lists of relevant articles to ensure that I captured all eligible studies. The implementation of the systematic review followed the *PRISMA* 2009 flow diagram (Figure 1).

### 2.2. Study Selection Criteria

The author included the studies in the review if they were implemented by *EBHI*, and concerned *CAC* technologies applied in a healthcare domain. More analytically, this review includes studies that: (a) focus on users/patients. (b) include the use of a context-awareness framework as a system related to patient health support; (c) contain evaluation data; and (d) monitor the performance of systems through clinical trials.

In the case of *clinicaltrials.gov*, appropriate clinical trials were initially sought using specific keywords. Then the author selected the publications (articles) related to these trials found at *clinicaltrials.gov*. Consequently, the author collected in the final collection only those containing a trial registration number.

Finally, the author removed from the final selection those that were study protocols and she excluded studies if they were not in English.

### 2.3. Screening Process

A two-stage review process the author performed, (a) initially excluding assignments based on the titles and their abstracts, and (b) then the remaining assignments based on the reading of the full text of the article.

One researcher reviewed the articles.

### 2.4. Data Extraction and Synthesis Strategy

One reviewer extracted information from the eligible studies into a data mining form, while two external independent evaluators examined the results for consistency and accuracy.

Using the *SF*/*HIT* to classify the findings of the review [13], the following information was collected: First author, Trial Registration Number, Publication year, Type of design, Impact by Type in accordance to *SF*/*HIT* framework (i.e., Preventive care, Adherence/Attendance, Efficiency, Perceived ease of use/Usefulness/feasibility, Effectiveness, Process of service delivery/ Performance, Safety/Privacy/Security, Acceptability, Cost effectiveness, Appropriateness and Satisfaction, Category of applied Information science in accordance to *MeSH* terms (i.e., Ambient Intelligence [L01.224.900.910.500], Health Information Systems [L01.313.500.750.300.361], Internet-Based Intervention [L01.224.230.110.500.688], Smartphone [L01.178.847.698.300.250], Virtual Reality [L01.224.160.875]/[L01.296.555], Telemetry [L01.178.847.675], Video Games [L01.224.900.930], Wearable Electronic Devices [E07.305.906], Mobile Applications [L01.224.900.685], Cloud Computing [L01.224.097]) and category of medical/health applications using *ICD-11* classification system (i.e., Factors influencing health status or contact with health services; Extension Codes (i.e., Home for the elderly, Life-style, Portable multi-parameter patient monitors and Outpatient clinic or health Centre); Endocrine, nutritional or metabolic diseases; Mental, behavioral or neurodevelopmental disorders; Diseases of the circulatory system; Neoplasms; Diseases of the nervous system; Diseases of the respiratory system; and Symptoms, signs or clinical findings, not elsewhere classified (i.e., Hemiplegia)).

The author grouped the information into the following categories: Impacts of studies by Type; Categories of applied Information science in accordance with *MeSH* terms and Categories of medical/health applications using the *ICD-11* classification system.

The mixed methods review was used in the study design are: (i) the guidelines for a scope proposed by Arksey and O’Malley [14], (ii) the Preferred Reporting Items for Systematic Reviews and Meta-Analyses statement [15] and (iii) the *SF/HIT* framework [13].

Moreover, the author applied a *Delphi* method [16] to improve the reliability of the study. Specifically, this study was given to two independent researchers for reading and then I discussed with them about the design and implementation of this study. We conducted three sessions using a teleconference tool. I considered the commentators’ comments in the final structure of the article.

The author depicts the analytical results of this above-described process in Table A1, Table A2 and Table A3 which show in detail the outcomes of this study.

## 3. Results

### 3.1. Retrieved Studies

The database search retrieved 342 citations. Considered 29 articles [17,18,19,20,21,22,23,24,25,26,27,28,29,30,31,32,33,34,35,36,37,38,39,40,41,42,43,44,45] that employ 26 trials in total, all published from 2014 until today (2022) (Table A1).

The trials belonged to one of the following categories in accordance with their study type: Randomized Clinical Trial, Non-Randomized Clinical Trial, Pilot study, Prospective Observational Pilot Study, Single Group Assignment Clinical Trial, Mixed Methods Study, Single-Subject Design Study, and a feasibility study (Table A1).

### 3.2. Descriptive Elements of the Studies

As far as *RQ1* and *RQ2* (Table A2) are concerned, the author presents systematically the descriptive elements of the studies and the types of *CAC*-based interventions according to the *ICD-11* classification system.

#### 3.2.1. 02—Neoplasms (Two Studies/Two Trials)

Time range of studies performed: 2019–2022.

The trials included in this domain aimed to determine whether people receiving chemotherapy for colorectal cancer are interested in participating in digital health physical activity interventions. Two pilot randomized controlled trials demonstrated that a remotely delivered health physical activity intervention that included a wristband for self-monitoring physical activity and *SMS* text messages is feasible and acceptable to colorectal cancer survivors [17,18]. The outcomes were adherence (e.g., Fitbit smartwatch wear and text response rate), acceptance and satisfaction with the digital intervention.

#### 3.2.2. 05—Endocrine, Nutritional, or Metabolic Diseases (Four Studies/Four Trials)

Time range of studies performed: 2011–2021.

The trials under this category are related to context-aware, mobile and web applications. Specifically, one trial [19] examined continuous self-*monitoring* by wearable technology with real-time feedback. This study proved that may be particularly useful to enhance lifestyle changes that promote weight loss in sedentary overweight or obese adults. Moreover, a study [20] focused on daily self-weighing as a self-monitoring strategy and shows promise for preventing weight gain in breast cancer survivors. The results of a trial study [21] showed that using a smartphone app and a smart band obtained beneficial results in weight loss in women and a reduction in body fat mass and percentage of body fat. Also, Ash et al. in [22] propose to use personalized big data from biosensors to substantiate human-delivered, client-centered exercise support for type 1 diabetes.

#### 3.2.3. 06—Mental, Behavioral, or Neurodevelopmental Disorders (Four Studies/Three Trials)

Time range of studies performed: 2011–2019.

The trial [23] is an intervention which based on mobile phone and Internet including ecological momentary intervention for unipolar depression and context sensing to identify mental health-related states. The application supporting architecture, in which machine learning models (i.e., learners) predicted patients’ mood, emotions, cognitive/motivational states, activities, environmental context, and social context based on concurrent phone sensor values (e.g., global positioning system, ambient light, recent calls). Moreover, the website included feedback graphs illustrating correlations between patients’ self-reported states, as well as didactics and tools teaching patients’ behavioral activation concepts. Moreover, a trial [24] studies experiential virtual scenarios with real-time monitoring for the management of psychological stress. Also, the trial [25] studies whether virtual reality is effective for exposure treatments. Using a virtual reality environment able to induce a high feeling of presence, cardiovascular and respiratory activities are monitored to evaluate both voluntary and autonomic effects of respiration on heart rate. The study analyzes both inter-beat intervals extracted from electrocardiogram and respiration (from a chest strip sensor). Another trial [26] examines the diagnostic evaluation of a smart tablet serious game to identify autism in children. The study is based on the acquisition of motion data of the child’s hand playing with the tablet. Movement data is acquired from the touch screen and the inertial movement unit sensor that detects the kinematics and contact forces of a gesture, respectively.

#### 3.2.4. 08—Diseases of the Nervous System (One Study/One Trial)

Time range of studies performed: 2020.

The trial [27] describes the development of a *context*-aware fall detection system based on inertial sensors and time of flight sensors that is robust to imbalance, which is trained and evaluated on real-world falls in people with Multiple Sclerosis. The system in its application achieved a sensitivity of 92.14% and a percentage of false-positive 0.65 false alarms per day.

#### 3.2.5. 11—Diseases of the Circulatory System (Three Studies/Two Trials)

Time range of studies performed: 2017–2021.

Two studies [28,29] using one clinical trial aimed to explore the feasibility of photoplethysmography-based smart devices for the detection of atrial fibrillation in real-world settings. Moreover, a comprehensive study, the so-called “*the box*” described in article [30] that examines the use of smart technology to improve outcomes in myocardial infarction patients. A box containing a weight scale, blood pressure monitor, activity tracker, and a wearable Electrocardiogram device supports the system. It offers patients the ability to connect their personal account with a physician’s account. The physician can then review the Electrocardiogram made by patients linked to their account, including the diagnosis given by the app’s algorithm and the symptoms reported by the patient.

#### 3.2.6. 12—Diseases of the Respiratory System (One Study/One Trial)

Time range of studies performed: 2020.

The objective of the study [31] was to investigate whether a system that uses a combination of measurements from respiratory *physiology* sensors can accurately assess asthma control in the home situation. This study showed that data acquired from home smart monitoring devices is strongly associated with the control of asthma.

#### 3.2.7. 21—Symptoms, Signs, or Clinical Findings Not Elsewhere Classified (One Study/One Trial)

Time range of studies performed: 2020.

The pilot study [32] confirmed the feasibility and applicability of the smart insole as a device to assess the *gait* of patients with hemiplegia due to stroke.

#### 3.2.8. 24—Factors Influencing Health Status or Contact with Health Services (Seven Studies/Six Trials)

Time range of studies performed: 2015–2021.

The study [33] evaluated the automated *mHealth* Intervention for physical activity promotion. Another pilot study [34] showed for the first time that monetary incentives for oral disease *management* are feasible, acceptable, and potentially efficacious in young children. The studies [35,36] aimed to assess the usability, perceived usefulness, and acceptance of the *mRehab* system by individuals with stroke and identify the challenges experienced by them when using the system remotely in a home-based setting. A feasibility study [37] examines the use of a technology-based system (i.e., a Microsoft Kinect camera and sensors) to motivate older adults in performing physical activity. Moreover, the trial [38] examines the use of a smartphone app to increase physical activity levels in insufficiently active adults. Also, a trial [39] conducted a theoretical analysis of stakeholder informed barriers and levers to the implementation of a novel exercise promotion m-health tool.

#### 3.2.9. X—Extension Codes (Six Studies/Six Trials)

Time range of studies performed: 2017–2022.

Intelligent personal assistants such as Amazon Echo and Google Home have become increasingly integrated into the home setting and, therefore, may facilitate behavior change via novel interactions. However, little is currently known about their potential role in this context. The feasibility study [40] aims to develop an Intelligent Personal Assistant Project and assess the acceptability and feasibility of this technology for promoting and maintaining physical activity and other health-related behaviors in both parents and children. The objective of the study [41] is to conduct an intervention trial that evaluates the feasibility of adding an electronic activity monitoring system to brief counseling within a primary care setting. Throughout the intervention, its feasibility and acceptability were evaluated, the change in primary outcomes (i.e., cardiovascular risk and physical activity measured by a *SenseWear* monitor), and the change in secondary outcomes (i.e., adherence, weight and body composition, health status, motivation, physical function, psychological feelings, and self-regulation). Also, a study [42] aims to evaluate the effectiveness of the advanced new generation Ecosystem service, which is developed to assist cardiac patients in adopting a healthy lifestyle and improving their quality of life. A pilot study [43] evaluates a new digital technology for remotely that has never been evaluated in ambulatory surgery. This study evaluated the use of a real-time remote monitoring device for outpatient surgery. This system enabled patients to record >60% of the required information. Also, a trial [44] was performed to investigate whether telemetric continuous glucose monitoring is associated with better glycemic outcomes and fewer patient health care worker contacts. Finally, the trial [45] demonstrated the feasibility of using smartwatch-based heart rate estimates to detect clenbuterol-induced changes during clinical trials.

### 3.3. Principal Findings

From what is known, this is the first review that examines the impacts of *CAC* interventions using *EBHI*. *CAC* is a very promising and recently introduced field in the applied sciences. Moreover, the combination of *CAC* via *EBHI* is a more complex and demanding issue and therefore has not yet been widely applied. This is evidenced by the fact that while no time limits were set for the selection of articles, all collected studies are published from 2011 onwards and in the appearance of *RCTs* (in 13 of 29 studies). Moreover, 15 of the 29 studies are published from 2019 onwards.

In the included studies, context was captured using sensors, biosensors, reports, *SMS*, voice, and video. Most studies used a combination of mobile apps, wearable and sensor technology, ambient intelligence, telemetry, and cloud computing to deliver personalized healthcare and interaction. All studies examined the impact of the interventions showing positive or neutral results, with a declining order of occurrence rate in perceived ease of use and feasibility, effectiveness, satisfaction, process of service delivery, acceptability, preventive care, and adherence.

It was also confirmed that *CAC* systems are a valuable and reliable method in health care provision in many different healthcare domains, e.g., home monitoring for the elderly (smart home), lifestyle, patient monitoring, and outpatient clinics or health centers, providing care to patients with endocrine, nutritional, or metabolic diseases, mental, behavioral, or neurodevelopmental disorders, with diseases of the circulatory, nervous, or respiratory systems, etc.

More specifically, as far as the *Health Information Technology* category is concerned, the analytical findings, with respect to RQ1, are (Table A2): 26 trials belong to *Ambient Intelligence* (100.00%), 15 trials to *Wearable Electronic Devices* (57.69%), 8 trials to *Mobile Applications* (30.77%), 8 to *Smartphone* (30.77%), 4 to *Health Information Systems* (15.38%) and to 3 *Virtual Reality* (11.54%). Fewer studies supported by *Internet-Based Intervention* (two trials–7.69%), *Telemetry* (two trials–7.69%), *Video Games* (one trial–3.85%), and *Cloud Computing* (3.85%) (Figure 2).

It is noted that the trials may belong to one or more of the *Health Information Technology* categories.

Of course, it was a prerequisite for this study that all trials be supported by ambient technology.

Also, as far as the medical/health applications category (by using the *ICD-11* classification system) is concerned, the analytical findings, with respect to *RQ2* follow (Table A2).

The most common categories of medical/health applications using *CAC* systems found in the literature that were tested via the *EBHI* belong to the following categories: (i) Factors influencing health status or contact with health services (seven studies/six trials); (ii) Extension Codes (i.e., Home for the elderly, Life-style, Portable multi-parameter patient monitors and Outpatient clinic or health Centre) (six studies/six trials); (iii) Endocrine, nutritional or metabolic diseases (four studies/four trials); (iv) Mental, behavioral or neurodevelopmental disorders (four studies/three trials); (v) Diseases of the circulatory system (three studies/two trials); (vi) Neoplasms (two studies/two trials); (vii) Diseases of the nervous system (one study/one trial); (viii) Diseases of the respiratory system (one study/one trial); (ix) Symptoms, signs or clinical findings, not elsewhere classified (i.e., Hemiplegia) (one study/one trial).

Moreover, the most cumulatively positive disease-specific impacts factors appear in descending order as follows (Figure 3): (i) Endocrine, nutritional or metabolic diseases; (ii) Factors influencing health status or contact with health services; (iii) Extension Codes (i.e., Home for the elderly, Life-style, Portable multi-parameter patient monitors and outpatient clinic or health center); (iv) Neoplasms; (v) Diseases of the circulatory system; (vi) Mental, behavioral or neurodevelopmental disorders; (vii) Symptoms, signs or clinical findings, not elsewhere classified (viii) Diseases of the nervous system; and (viii) Diseases of the respiratory system.

Regarding *RQ3* (Table A3) the studies showed positive results in relation to: (i) perceived ease of use/usefulness/feasibility (65.38%); (ii) effectiveness (46.15%); (iii) satisfaction (34.62%); (iv) process of service delivery/ performance (30.77%); (v) acceptability (30.77%); (vi) preventive care (19.23%); (vii) adherence/attendance (19.23%); (viii) safety/privacy/security (11.54%). No positive or negative results have been observed in efficiency, cost effectiveness and appropriateness.

Thus, the most cumulative positive results for all diseases per impact follow in descending order (Figure 4): firstly perceived ease of use/usefulness/feasibility appears, secondly the effectiveness and thirdly the users’ satisfaction. Obviously, the study of efficiency, cost-effectiveness and appropriateness are a more specialized, costly, and time-consuming process and that is why these results have not yet appeared.

### 3.4. Comparison with Previous Literature

On the one hand, in a previous relatively similar review [13] that concerns the application of all technologies in *EBHI*, lower levels of positive results appeared in cost-effectiveness (only one out of six trials—16.7%) and efficiency (one out of three trials—33.3%) are depicted. On the other, in this study, no positive or negative results have been observed in efficiency, cost-effectiveness, and appropriateness. The fact that there are no cost-effectiveness studies regarding the application of *HIT* and more especially in *CAC* technologies in health domain explains their non-entry into widespread use.

Another quite similar review, which took place in 2019, did not identify *RCTs* to evaluate the efficacy of a context-aware system [46]. But in the study presented here, 13 *RCTs* were collected from a total of 29. The application of *RCTs*, as already described in the Introduction, ensures that the data and impacts are more reliable and accurate. As a result, increasingly reliable measurements were obtained regarding the implementation of *CAC* systems. More specifically, the efficacy (/effectiveness) was measured and gave a positive impact in 12 trials (46.15%).

The review [47] also examines some key technical issues as it focuses on common design and implementation patterns of intelligent *IoT*, sensor-based, smartphone-based, and microcontroller-based healthcare systems.

The authors of the article [48] surveyed previous work and presented the challenges and future directions for effectively learning context-aware rules from smartphone data. This survey focused on modeling techniques, e.g., time-series modeling, contextual rule discovery by considering multi-dimensional contexts, such as temporal, spatial, or social contexts, and incremental learning to dynamic updating of rules. The applications that the authors study concern not only e-health services such as this review, but also transportation services, city governance, industry, e-commerce, context-aware cyber-security intelligence, and smart city services. For this reason, after all, they are not related to clinical studies or *RCTs*.

A comprehensive review [49] explore the state-of-the-art smart healthcare systems. This study does not focus on context-aware systems, but generally examines the most significant areas of research. It includes wearable and smartphone-based health monitoring, machine learning for predictive analytics, and assistive frameworks developed for assisted living environments, and social robots. Thus, this review provides a holistic overview and comparison of state-of-the-art research, but no clinical trials of the developed frameworks were identified in this.

Finally, Islam et al. [50] conducted a literature review specializing in human activity using tools of convolutional neural networks.

### 3.5. Strengths and Limitations

In this study the author conducted a review, to examine the effectiveness of healthcare interventions via *EBHI*. Thus, applies a sufficiently clear, systematic, and thorough search strategy over multiple clinical trial registries and academic digital libraries. Moreover, the application of *EBHI* is highly valuable, improves transparency, and emphasizes the importance of empirical evidence over preconceived knowledge. Thus, the selection of clinical trials ensures that their use enhances the quantitative and/or qualitative characteristics of research to create a strong, empirically derived response to a focused research question.

More specifically, this review has many strengths in relation to the design of the study.

Firstly, the author performed an extensive search in several databases to ensure that the author captured all relevant studies.

Secondly, articles were searched in the largest clinical trial registration database i.e., *clinicaltrials.gov*.

Thirdly, the author performed the search procedures more than once, and she evaluated many alternative search criteria i.e., she evaluated and applied many different keyword combinations on many online databases.

The results of this study, however, are subject to some serious limitations.

Firstly, only one researcher conducted this study. However, the author applied the *Delphi* method [16] to improve the quality characteristics of the study. Specifically, she gave the article for reading to two independent researchers and then we discussed the design and implementation of this study. Subsequently, she considered the commentators’ comments in the final structure of the article.

Secondly, since this study deals with two emerging fields, *EBHI* and the implementation of *CAC*, there is a lack of timeless and experimental studies, which prevents the evaluation of the impacts of these interventions. Therefore, a review has taken place instead of a systematic review.

Thirdly, the final selection of clinical trials included in this review limited in some respects the strength of this study. The weakness is due mainly to issues related to the lack of mapping of the technological infrastructures. This takes place because many of the studies included in this review did not even describe the technical framework or demonstrated that they are supported by an overly simple and rudimentary technological structure. However, they were included in this study as they were deemed appropriate because they describe the impacts and outcomes that were a prerequisite for inclusion in this study.

Finally, another limitation was the exclusion of non-English documents.

## 4. Discussion

Initially, the author did not set a time limit on the search criteria of this study. This is because it was expected that *CAC* technology in *EBHI* would be very recent. Although, few were found even in an earlier time (only two from 2011). These are two simple mobile applications, the first of which is related to diet- and physical activity-based lifestyle interventions, and the second is an intervention that aims to identify mental health-related states and depression.

More specifically, from what the author described in the previous section, the range of diseases that have been studied so far is limited. These studies focused mainly on nutritional and metabolic diseases, on continuous care at home, outpatient, and lifestyle support. However, it is estimated that in the future the list of diseases to be supported by *CAC* systems on *EBHI* will be significantly expanded.

Moreover, the most acceptable and satisfactory systems by users are related to endocrine, nutritional, or metabolic diseases. Obviously, the development of these systems is simple, their use is convenient for users and concerns a large portion of the population, mainly young and digitally educated.

But wider and more integrated studies are crucial to focus mainly on issues concerning: (i) management in matters related to a larger number of diseases and health issues (ii) more accurate and detailed mapping of the technological infrastructures used and (iii) measurements in relation to more impacts, especially in cost-effectiveness. As a result, it is expected to significantly improve the research and the applications in the sector of *CAC* in *EBHI*.

## 5. Future Directions and Challenges

Nowadays people live longer than ever before, and the world is leading in a continually aging society. At the same time, the states are looking for improving the quality of their citizens’ health, as well as to reduce the cost of providing health services. For these reasons, today digital health is an emerging dynamic market with enormous potential that will change the applied health standards worldwide.

Also, today all the data related to healthcare are being digitized. Most hospitals use digital systems. In addition, the development of information technology has enabled the creation of a significant amount of health data generated by the patient in his daily life. This data is collected using mobile devices and mobile health applications to manage health conditions, including the management of chronic diseases or the immediate provision of care.

Thus, challenges should focus on careful evaluation of these modern applied health technologies such as *CAC* systems for performance, cost-effectiveness, use of standards, benchmarking (e.g., meta-analysis), reliability, security, and compliance with ethics.

Therefore, although the implementation of clinical trials and more specifically *RCTs* is a time-consuming and expensive method, it is considered the “*gold standard*” and guarantees the most reliable and robust evaluation of these modern technologies. Hence, the future direction for achieving the evaluation of these systems is estimated to focus on the use of clinical trials and *RCTs*. Thus, before long it is expected that the research community will use the combined application of *EBHI* and *CAC* systems more widely.

In this study, the use of accomplished clinical trials and *RCTs* (i.e., the *EBHI* method) has already made its appearance in the evaluation of *CAC* in health and the impacts of these systems have been presented.

However, future similar but broader research that focuses on capturing the technological framework, even if the research includes incomplete studies (e.g., clinical protocols, studies in the phase of recruitment or implementation) will contribute positively and will significantly promote the research in this field and its finest implementation. Thus, such an exploration may indicate the design of technological architectures and *IT* solutions for health that will be considered the most appropriate and advantageous in the field under study (i.e., *EBHI* and *CAC* technologies).

## 6. Conclusions

The use of context-aware systems in healthcare is a promising new area of research. The evaluation of these systems with the application of *EBHI* provided clarified and reliable impacts in terms of self-management practices and management of factors influencing health status, lifestyle improvement, and management of long-term diseases (i.e., endocrine, nutritional, or metabolic diseases, mental, behavioral, or neurodevelopmental disorders, and diseases of the circulatory, nervous, or respiratory system).

Improvements in healthcare provision using *EBHI* were observed in the findings of *CAC* application, i.e., perceived ease of use and feasibility, effectiveness, satisfaction, process of service delivery, acceptability, preventive care, and adherence.

Future studies should clearly indicate the intervention and the areas of health to which the intervention is applied following standards such as *ICD11* and *MeSH* terminologies.

Finally, studies should provide data, in addition to what has been reported so far, on those related to efficiency, cost-effectiveness, and appropriateness of the trials.

## Figures and Tables

**Figure 1 healthcare-10-00685-f001:**
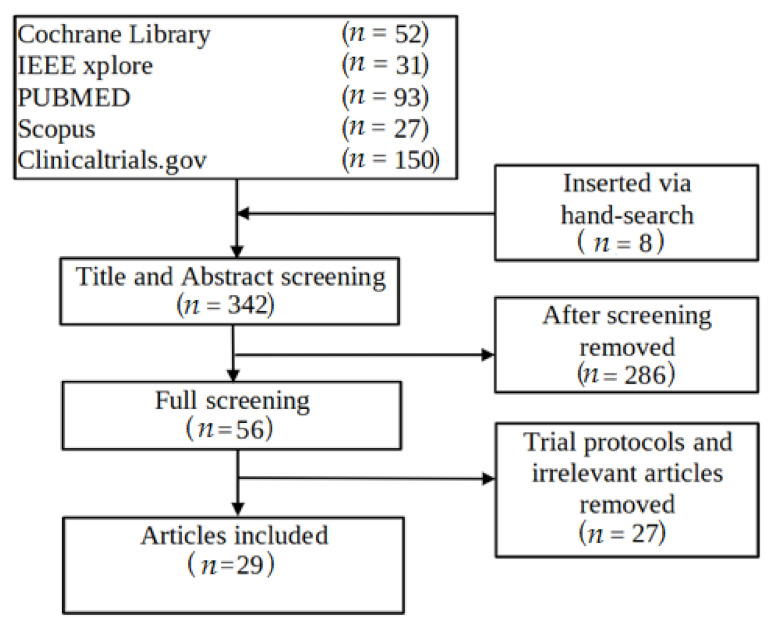
*PRISMA 2009* flow diagram of included studies.

**Figure 2 healthcare-10-00685-f002:**
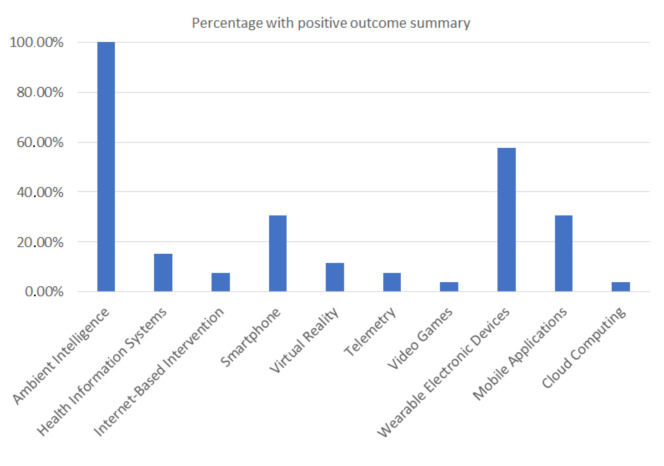
The percentages with positive values of the most common categories of applied information science (in accordance with *MeSH* terms) using *CAC* systems.

**Figure 3 healthcare-10-00685-f003:**
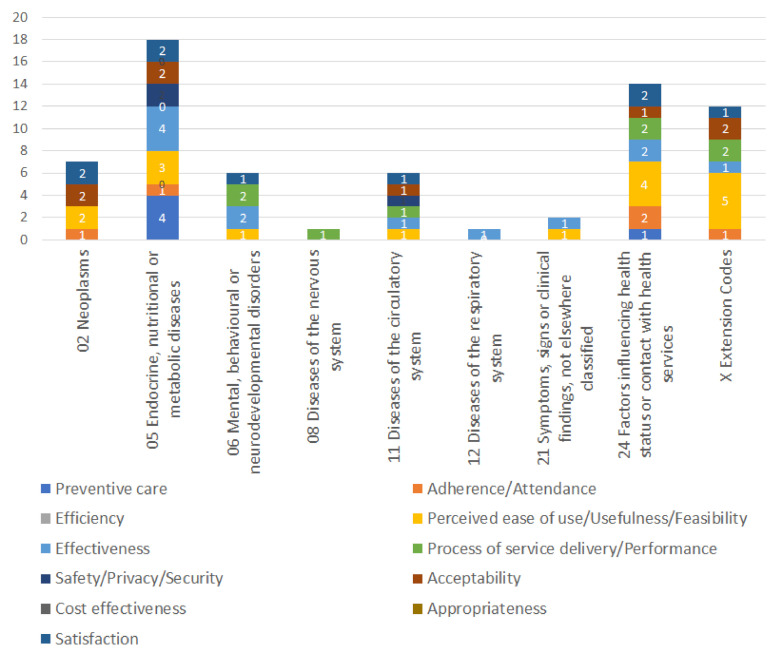
Correlation of disease-specific factors (i.e., *ICD-11*) with the impacts of *CAC* systems on *EBHI*.

**Figure 4 healthcare-10-00685-f004:**
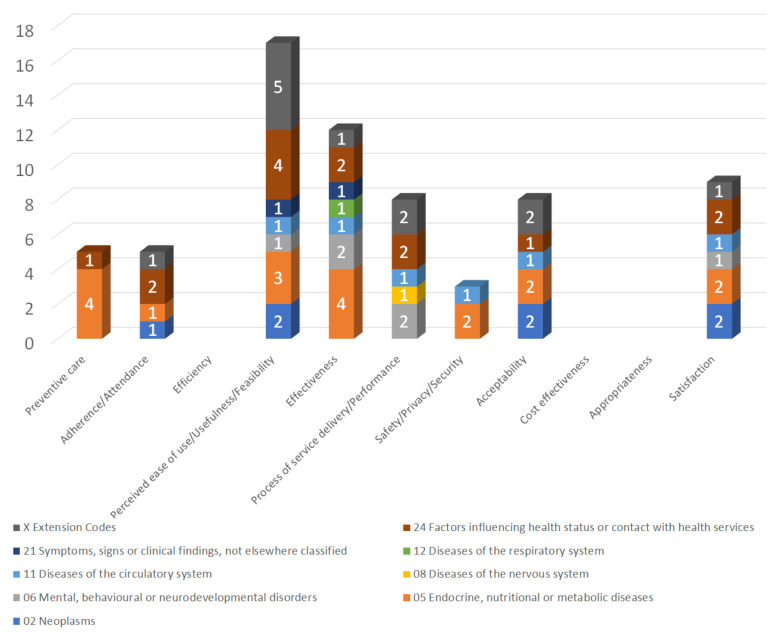
The percentages of the impacts of the studies/trials on Context Aware Systems in *EBHI*.

## Data Availability

Data from this research are not available elsewhere. Please contact the author for more information, if required.

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
