# Peer review of "Impacts on Context Aware Systems in Evidence-Based Health Informatics: A Review"

_healthcare, 2022, doi:10.3390/healthcare10040685_

Round 1

Reviewer 1 Report

Small language mistakes. In some cases, the sentences are too long and meaningless.

Author Response

First of all, I thank the reviewer for his/her important and good comments. Next, I answer about the corrections / improvements I made in manuscript following the reviewer’s comments / syggestions.

Moreover, I quote the manuscript where I track the changes in detail.

Point 1: Small language mistakes. In some cases, the sentences are too long and meaningless.

Response 1: Yes, I searched the whole manuscript and Ι corrected it. I hope I got it all right. In some cases I shortened the sentences to better convey their meaning.

Reviewer 2 Report

Segmenting the introduction part has reduced the fluency of the writing.
It is suggested to specify the discussion section using "discussion" word.

Author Response

First of all, I thank the reviewer for his/her detailed and valuable comments. Next, I answer about the corrections / improvements I made in the manusctipt following the reviewer’s comments / syggestions.

Moreover, I quote the manuscript where I track the changes in detail.

Point 1:  Segmenting the introduction part has reduced the fluency of the writing.
Response 1: It was really a problem to split the introduction into two sections. I finally managed to unify it. I think it became much better.

Point 2:  It is suggested to specify the discussion section using "discussion" word.

Response 2: I added the section: “Discussion” in the manuscript.

Reviewer 3 Report

In my opinion, the paper is well written and has good technical components and clearly described but a rewrite is required before accept. I have some suggestions and questions.

Comment #1: Firstly, for Abstract section, it is lengthy and it should be refined to precisely illustrate what authors have done in this paper within 200 words.

Comment #2: The review is in a shallow and abstract form. In my opinion it can provide more details. In other words, the current form focuses more on the results than their innovations.

Comment #3: The proposed method matches with some recently published article. Authors should mention a comparison of proposed work with existing reviews.

Comment #4: There are lots of typos. English needs to revise again with a professional editing service. Also, the figures are not clear in some cases.

Comment #5: Kindly write the research gap and future prospects of this study. Provide the future directions and challenges in a tabular form.

Comment #6: The discussion section should be more detailed and expanded. In its current form, the discussion section is a summary of the 'results' showing the performance parameters, which are themselves derived from the systems examined.

Comment #7: Some recent healthcare platforms must be cited from the followings to properly present the background of the study. https://arxiv.org/abs/2202.03274, https://ieeexplore.ieee.org/abstract/document/9565155, https://doi.org/10.18280/ria.330605

Author Response

First of all, I thank the reviewer for his/her detailed and valuable comments. Next, I answer point by point about the corrections / improvements I made within the text of the article following the reviewer’s comments / syggestions. Moreover, I quote the manuscript where I track the changes in detail.

Point 1: In my opinion, the paper is well written and has good technical components and clearly described but a rewrite is required before accept. I have some suggestions and questions.

Firstly, for Abstract section, it is lengthy and it should be refined to precisely illustrate what authors have done in this paper within 200 words.

Response 1: Ιnitially following the reviewers’ instructions I corrected the whole text so that it appears in the third person (e.g. as “the author” or “she”) and nowhere in a passive voice or first person. 

I also modified the abstract in some points to make the auhor’s contribution in this article clearer (eg . “The author included the articles in the review if they implemented….”, “Τhe author noticed improvements…”, “She also confirmed that CAC systems are a valuable and reliable method…”, “The author presented that the evaluation of CAC systems in EBHI presents positive effects..”, “These implications are presented in this article in a detailed, clear, and reliable manner”)

But, as for the limit of words in abstart I managed to be 215 words. I hope this is acceptable.

Point 2:

Comment #2: The review is in a shallow and abstract form. In my opinion it can provide more details. In other words, the current form focuses more on the results than their innovations.

Response 2: Indeed, the initial design aimed to select the articles and then to present in detail and comparatively a) the technological architecture and innovation and b) the impacts and benefits of the implementation of these sytems. However, two reasons forced me to divide my study into two parts and to present this in two different articles (The first part is presented in this article).

  1. Clinical protocols and clinical trials that have not been completed should be removed from the initial collection of articles in order to study the relative impacts (There are not exist impacts in clinical trials). On the contrary, technological architecture and innovation can be cited even if it is presented in a clinical protocol and the clinical trial has not yet been done. Also in many cases integrated clinical trials present their impacts but do not describe their architectural structure and innovation. Thus these two parts of my study while have the same initial set of articles the final inclusion of them is significantly differentiated.
  2. Based on the above, the article would be too long if I included both parts of the study in one article. In addition, this will significantly confuse and tire the reader. I hope the next article will cover this weakness.

Point 3: The proposed method matches with some recently published article. Authors should mention a comparison of proposed work with existing reviews.

Response 3:. I compare this study with previous literature in section 3.4. More specifically I present two similar studies. The first study concerns the application of all technologies in EBHI and is more general. The second study which is quite similar review with the present onebut it did not identify RCTs to evaluate the efficacy of a context-aware system. Οn the contrary in the study presented here, 13 RCTs were collected from a total of 29. Clearly the reliability of a study is enhanced by the application of RCTs. This fact is mentioned in the introduction (i.e. “The research and medical community have widely accepted the conduct of RCTs and consider it the ‘gold standard’, second in the ranking after systematic review. This is because RCTs provide a strict standard that clearly defines the design, conduct, monitoring, control, recording, analysis and reporting of clinical trials which ensures that data and outputs are reliable and accurate while protecting the rights, integrity and confidentiality of those involved…”).

Moreover I added in the section the following sentence: “…The application of RCTs, as already described in the Introduction, ensures that the data and impacts are more reliable and accurate”.

Detailed data of the studies and the comparison with the present one are described in section 3.4.

Also I added four reviews in the related work.     Three of them are the ones you mentioned in point 7.

Point 4: There are lots of typos. English needs to revise again with a professional editing service. Also, the figures are not clear in some cases.

Response 4: The text was thoroughly checked for typographical errors and errors of English spelling. I also improved the figs so that they are more legible.

Point 5: Kindly write the research gap and future prospects of this study. Provide the future directions and challenges in a tabular form.

Response 5: Yes, I added the section: “Discussion” in the manuscript. I hope its ok.

Point 6: The discussion section should be more detailed and expanded. In its current form, the discussion section is a summary of the 'results' showing the performance parameters, which are themselves derived from the systems examined.

Response 6: I added the section: “5. Future directions and challenges” in the manuscript.

Point 7: Some recent healthcare platforms must be cited from the followings to properly present the background of the study. https://arxiv.org/abs/2202.03274, https://ieeexplore.ieee.org/abstract/document/9565155, https://doi.org/10.18280/ria.330605

Response 7: Thank you. The articles were relevant and very useful. I added them to the manuscript  as sources and references.

Reviewer 4 Report

The abstract is well written and the reader can determine the problem and how the SLR was undertaken. The introduction does a good job of setting the context. All key terms are defined. In section 1.2 first paragraph several definitions require references. The research questions are appropriate for this work, however when you commence with "what", you get descriptive question, try to make your research question complex and analytical, begin with "how" or "why". There are a number of frameworks that can assist  you, e.g., the PEO framework.

The methodology used is sound. PRISMA is a well known and well respected approach. This is good work. 29 papers is just on the borderline of being a sufficient sample. Also it is better if you can focus on recent publications. I think 2014 is a little old, however if you can include an argument as to why papers more than 5 years old should be included this would be ok.

Try to avoid writing in the first person, e.g., section 3.2 p.6 first paragraph.

Section 3.2 is a strength of the paper, the classifications are clearly described. Section 3.3 circles back to the RQ and provides a short robust discussion around each RQ. The author links the classifications identified in section 3.2 to the RQ's in section 3.4. A deeper analysis on their impact and the practical and theoretical implications would be good.

The appendices are good,  however table A3 is hard to read.

Overall a nicely written, interesting paper. 

Author Response

First of all, I thank the reviewer for her/his detailed and valuable comments. Next, I answer point by point about the corrections / improvements I made in the manuscrip following the reviewer’s comments / syggestions.

Point 1: The abstract is well written and the reader can determine the problem and how the SLR was undertaken. The introduction does a good job of setting the context.

All key terms are defined. In section 1.2 first paragraph several definitions require references.

Response 1: Yes, I put the nessesary references of key terms as footnotes by using MesH browser (i.e. According to the US National Library of Medicine's MeSH).

Point 2: The research questions are appropriate for this work, however when you commence with "what", you get descriptive question, try to make your research question complex and analytical, begin with "how" or "why". There are a number of frameworks that can assist  you, e.g., the PEO framework.

Response 2: Yes, I tried to correct them. I managed the research questions (formulating research questions and answers) in a more elaborate and appropriate manner. Moreover, I quote the manuscript where I track the changes in detail.

Point 3: The methodology used is sound. PRISMA is a well known and well respected approach. This is good work. 29 papers is just on the borderline of being a sufficient sample. Also it is better if you can focus on recent publications. I think 2014 is a little old, however if you can include an argument as to why papers more than 5 years old should be included this would be ok.

Response 3: Initially, I did not set a time limit on the search criteria of this study. This is because it was expected that CAC technology in combination with EBHI would be very recent. Finally, the studies that were identified were minimal and most of them were recent. Few were found even in earlier time (only 2 from 2011). However, I thought I would extract useful data and I kept them.

Initially, the I did not set a time limit on the search criteria of this study. This is because it was expected that CAC technology in EBHI would be very recent. Although, few were found even in earlier time (only 2 from 2011). (I thought I would extract useful data and I kept them). These are two simple mobile applications which the first is related to diet-and physical activity-based lifestyle intervention and the second is an intervention that aims to identify mental health-related states and depression.

I put these findings, among other things, to the new section that I added i.e. “Discussion”

Point 4: Try to avoid writing in the first person, e.g., section 3.2 p.6 first paragraph.

Response 4: Yes, of course. I corrected the fisrt person and the passive voice anywhere in the manuscript. I replace them with the third person (i.e. the author…)

Point 5: Section 3.2 is a strength of the paper, the classifications are clearly described. Section 3.3 circles back to the RQ and provides a short robust discussion around each RQ. The author links the classifications identified in section 3.2 to the RQ's in section 3.4. A deeper analysis on their impact and the practical and theoretical implications would be good.

Response 5: I added 3 figures and some relative text to better convey the meaning of the findings. Also I added the sections : Discussion and Future directions and challenges. Moreover, I quote the manuscript where I track the changes in detail. Τhere you can see the dispersed changes I made to the manuscript to improve it semantically. I hope I did it.

Point 6: The appendices are good, however table A3 is hard to read. Overall a nicely written, interesting paper. 

Response 6: Thank you. In fact, I corrected tables A2 and A3 to make them more legible.

Round 2

Reviewer 3 Report

Well revised. Good work!